

# Fine grain emotion analysis in Spanish using linguistic features and transformers

Alejandro Salmerón-Ríos,  José Antonio García-Díaz,  Ronghao Pan and Rafael Valencia-García

Departamento de Informática y Sistemas, Universidad de Murcia, Campus de Espinardo, Murcia, Murcia, Spain

## ABSTRACT

Mental health issues are a global concern, with a particular focus on the rise of depression. Depression affects millions of people worldwide and is a leading cause of suicide, particularly among young people. Recent surveys indicate an increase in cases of depression during the COVID-19 pandemic, which affected approximately 5.4% of the population in Spain in 2020. Social media platforms such as X (formerly Twitter) have become important hubs for health information as more people turn to these platforms to share their struggles and seek emotional support. Researchers have discovered a link between emotions and mental illnesses such as depression. This correlation provides a valuable opportunity for automated analysis of social media data to detect changes in mental health status that might otherwise go unnoticed, thus preventing more serious health consequences. Therefore, this research explores the field of emotion analysis in Spanish towards mental disorders. There are two contributions in this area. On the one hand, the compilation, translation, evaluation and correction of a novel dataset composed of a mixture of other existing datasets in the bibliography. This dataset compares a total of 16 emotions, with an emphasis on negative emotions. On the other hand, the in-depth evaluation of this novel dataset with several state-of-the-art transformers based on encoder-only and encoder-decoder architectures. The analysis compromises monolingual, multilingual and distilled models as well as feature integration techniques. The best results are obtained with the encoder-only MarIA model, with a macro-average F1 score of 60.4771%.

# INTRODUCTION

Mental health is a major global public health problem. Specifically, depression affects approximately 300 million people worldwide and is a major contributor to suicide; indeed, depression is the third leading cause of death among people aged 10 to 24. According to the National Statistics Institute (https://www.ine.es/) (INE) and the latest European Mental Health Survey conducted in Spain, an increase in cases of depression in the population due to the pandemic was observed between July 2019 and July 2020 (*World Health Organization (WHO), 2022*). In 2020, a total of 5.4% of the population (about 2.1 million people) experienced some form of depression. Early diagnosis of mental health

Corresponding author
José Antonio García-Díaz,
joseantonio.garcia8@um.es

problems is critical to effective treatment, as individuals are typically reluctant to seek help from specialized clinicians to treat their conditions. However, social media platforms are often used by these individuals to discuss their difficulties and find emotional support. This presents a significant opportunity for automated analysis of social media data to detect potential changes in their mental state that would otherwise go unnoticed.

Emotion analysis (EA) is an aspect of automatic document classification (ADC) that attempts to identify emotions conveyed in text documents. Ongoing research and development in this area is motivated by the growing awareness of mental health, the increased use of social networks, and the need to identify users' states of mind. Advances in natural language processing (NLP) research have led to the development of innovative techniques that show remarkable performance in tasks such as EA, as demonstrated by Transformer-based models (*Acheampong, Nunoo-Mensah & Chen, 2021*; *García-Díaz, Colomo-Palacios & Valencia-García, 2021*). This interest in NLP has led to exciting advances in the field.

One way to detect mental disorders is through everyday communication. Research conducted in *De Choudhury et al. (2014)*; *Guntuku et al. (2017)* has shown that individuals with mental disorders exhibit variations in language and behavior, including increased expression of negative emotions and self-absorption. As a result, clues to an individual's altered mental state can be found in their online posts. Thus, text emotion recognition is an important aspect of natural language, where emotions in written text are identified using existing emotion-tagged datasets and algorithms. Unlike Sentiment Analysis (SA), which generally categorizes text into broad classes (*Medhat, Hassan & Korashy, 2014*), EA categorizes text into fine-grained emotional scales. The evaluation of emotions is a well-researched topic, and the scales typically include Paul Ekman's six basic emotions (*Ekman & Davidson, 1993*). However, there is limited attention to the Spanish language in the literature. In addition, the majority of studies only address the basic emotions outlined by Ekman, leaving out other emotions that can help identify signs of depression. However, considering complex emotions such as sadness, loneliness, and hopelessness may improve early detection of depression.

For the purposes of this article, we have defined the following research questions related to EA for the detection of mental disorders:

- **RQ1.** How reliable is the identification of negative fine-grained emotions?
- **RQ2.** What is the best approach to face emotion analysis using text as input?
- **RQ3.** Are generative models effective at identifying different emotions?

The article makes two significant contributions to the field of EA for the detection of mental disorders: (1) The dataset we compile and analyze includes 16 different emotions using a multi-classification scheme. This dataset provides a unique approach by including emotions and states beyond those defined by Ekman's basic emotions, including emotions such as loneliness, depression, suicidal ideation, and hopelessness. (2) We evaluate this dataset with several encoder-only, encoder–decoder, and feature integration models for text generation in EA due to the promising results reported in *Plaza-del Arco, Nozza & Hovy (2023)*.

The article is organized as follows: 'State of the art' presents a summary of the current literature on NLP and EA, as well as the state of the art of EA in mental disorders. 'Materials and methods' describes the materials and methods used in this study, including a detailed description of our proposed pipeline. 'Dataset' describes the compiled dataset and presents the experiments for evaluating the dataset. Next, 'Results and analysis' illustrates the experiments conducted to evaluate the different strategies analyzed and discusses the results. Finally, 'Conclusions and further work', presents the implications of the results and possible future work.

## STATE OF THE ART

This section contains a review of recent research and literature on the use of NLP techniques for analyzing emotions (see 'Natural language processing for emotion analysis') and their application in the medical field (see 'Emotion analysis in the medical domain').

### Natural language processing for emotion analysis

EA serves as a tool for identifying specific human emotions, including anger, sadness, and fear, among others. With the advancement of Internet services, people are increasingly using social media platforms to express their emotions, participate in discussions, and exchange views on a variety of topics. In addition, some users provide feedback and review various products and services on e-commerce websites. In today's digital age, organizations across all industries are experiencing a digital revolution, resulting in a significant increase in both structured and unstructured data. A critical activity for these organizations is to transform unstructured data into valuable insights that can inform the decision-making process (*Munezero et al., 2014*). Identifying emotions from user-generated text enables the recognition of their emotional state and perspective on products or services. As a result, vendors and service providers are inspired to improve their current systems, products or services.

Despite the latest advances in NLP, EA from text remains a challenging task due to ambiguities and the introduction of new slang or terminology.

Broadly speaking, emotion models can be divided into two categories:

- **Dimensional model of emotions**: This model represents emotions using three parameters: (1) valence, which refers to the polarity of an emotion; (2) arousal, which measures the level of intensity associated with a feeling; and (3) power (also known as dominance), which refers to the degree of control over an emotion.
- **Categorical model of emotions**: This model represents emotions as discrete parameters, such as sadness, happiness, and anger, among others. Emotions are categorized into varying number of groups, ranging from four to eight depending on the specific model. In this category, most researchers prefer Ekman's or Plutchik's (*Plutchik, 1980*) emotion models as a fundamental basis. These models define emotional states to be used when labeling sentences or documents. For example, Ekman's six basic emotions have been used by *Batbaatar, Li & Ryu (2019)*, *Becker, Moreira & dos Santos (2017)*, while some researchers have introduced one or two additional emotional states, such as

Plutchik's (*Mohsin & Beltiukov, 2019*; *Park, Bae & Cheong, 2020*) to develop customized emotion models. In addition, in existing studies or cases, researchers have used a classification system that includes more than 10 emotions. For example, *Cowen & Keltner (2017)*, employed a self-report measure that captured 27 different categories of emotion, bridged by a continuous gradient. Similarly, *Lazarus & Lazarus (1994)* considered emotional analysis as a set of 15 emotions.

Text-based EA is usually considered as a supervised machine learning (ML) classification problem, which involves assigning one or more emotion categories to documents. The common pipeline in text-based EA involve steps such as (1) data preprocessing, (2) tokenization and lemmatization, (3) feature extraction, and (4) machine learning algorithms *Saffar, Mann & Ofoghi (2023)*. The first step is to preprocess textual data. On social media platforms, individuals often express their emotions informally, resulting in highly unstructured data that is difficult for machines to extract meaningful information from. Therefore, this step plays a key role in ensuring data quality, as it has a profound impact on subsequent analysis. The second stage is tokenization and lemmatization, which involves tokenizing the text into sentences and words. These tokens are then transformed into understandable numerical vectors suitable for ML algorithms. The third stage is feature extraction; some prominent examples include techniques such as Bag of Words (BoW), term frequency-inverse document frequency (TF–IDF), and word embedding. The fourth stage involves training a supervised ML algorithm on the extracted features to identify the emotions present in the text.

Traditionally, first text-based EA systems rely on approaches that use affective lexicons associated with psychological states (*Murthy & Kumar, 2021*) such as WordNet-Affect (*Strapparava & Valitutti, 2004*). WordNet-Affect is an extended form of WordNet (*Miller, 1995*) that includes affective words annotated with emotion labels. However, lexicon-based methods face challenges such as keyword ambiguity and limited linguistic information. Another popular approach is to train ML-based models such as support vector machine (SVM) or decision trees from statistical features such as TF–IDF or Bag of Words (BoW). Modern approaches are based on word embeddings, which rely on obtaining word or sentence representations to identify emotions in text. Word2Vec (*Mikolov et al., 2013*), GloVe (*Pennington, Socher & Manning, 2014*), fastText (*Joulin et al., 2017*), and ELMo (*Peters et al., 2018*) are commonly used embedding techniques. The state of the art for word or sentence embeddings is the use of Transformers architecture, as it provides contextual word embeddings thanks to attention mechanisms. Moreover, Transformers approaches provide transfer learning, which allows to pre-train a model with generic unsupervised tasks and then fine-tune it to specific tasks where we have fewer training instances (*Nandwani & Verma, 2021*). Moreover, all these methods can be combined in hybrid approaches that overcome the limitations of each approach separately.

Although EA is a frequently studied topic with a considerable amount of literature available, little attention has been paid to the Spanish language. One study that has addressed this issue is the EmoEvalEs shared task (*Plaza-del Arco et al., 2021*) organized by IberLEF 2021, which aims to promote the recognition and evaluation of emotions in the Spanish

language. Our research group participated in this task (*García-Díaz, Colomo-Palacios & Valencia-García, 2021*), where we presented a method that merges explainable linguistic features with BETO (*Cañete et al., 2020*), resulting in an accuracy rate of 68.5990%.

## Emotion analysis in the medical domain

Mental disorders alter the way a person feels, thinks, or acts and is a major public health problem, causing disability and poor well-being worldwide (*Rehm & Shield, 2019*). The World Health Organization (WHO) highlights that one in eight people struggle with mental disorders, posing a significant economic challenge to governments (https://www.who.int/news-room/fact-sheets/detail/mental-disorders). The COVID-19 pandemic has exacerbated this problem and increased its impact (*Skaik & Inkpen, 2020*). Early detection of mental illness can prevent its progression to a severe state and allow for intervention (*Leiva & Freire, 2017*). However, most patients with mental disorders do not receive effective diagnosis and treatment due to ignorance about mental health assessment and the stigma associated with these illnesses.

Social media platforms such as X (formerly known as Twitter) have become valuable sources of information for the healthcare industry, as more and more people turn to these platforms to share their ailments and seek emotional support. This presents an important opportunity for automated processing of social media data to detect changes in mental health status that might otherwise go unnoticed, before they develop into more serious health consequences (*Zhang et al., 2023a*). It is therefore imperative to detect mental disorders in individuals at an early stage, as this could have life-saving implications. NLP is playing an increasingly important role in the processing of social media data and has been used to facilitate tasks such as Sentiment and Emotion Analysis and mental health assessment.

The CLEF eRisk Lab (https://erisk.irlab.org/) has been held annually since 2017 as a shared task aimed at identifying early signs of mental disorders from social media posts. The 2023 edition of eRisk (*Parapar et al., 2023*) focused on detecting early signs of depression, pathological gambling, and measuring the severity of eating disorder symptoms. In the working note published by our research group on MentalRisk (*Pan, García-Díaz & Valencia-García, 2023*), we achieved good performance of pre-trained models based on Transformers for the detection of mental disorders. Furthermore, in another study (*Pan, Díaz & Valencia-García, 2023*), we observed that emotions are a feature that complements depression detection models and could help improve their performance.

In addition, previous computational studies have shown that individuals with mental disorders consistently exhibit changes in their speech and behavior, including an increased prevalence of negative emotions and a heightened focus on self-attention (*Guntuku et al., 2017*). As a result, daily communication is essential in detecting mental disorders. To protect patients from mental health problems such as depression, physicians should use automated sentiment and emotion analysis, as suggested in *Singh, Jakhar & Pandey (2021)*.

There has been a growing focus on emotional dysfunction as central to depression, and thus there has been much research on the relationship between mental disorders and emotional variability. For example, *Rottenberg (2017)* reports on what is known about how

depression affects emotional reactivity and regulation, while acknowledging the vast areas that remain unknown. Some of the emotions suggested by the authors are directly related to mental disorders, like hopelessness or sadness.

Given that emotions are an important part of human nature and can affect people's behavior and mental states (*Canales & Martínez-Barco, 2014*), a correlation has been found between emotions and mental disorders such as depression (*Compare et al., 2014*). For example, in *Joormann & Gotlib (2010)*, it was shown that the severity of depressive symptoms is associated with an increasingly inverse relationship between positive and negative emotions. Furthermore, in *Dejonckheere et al. (2019)*, *Dejonckheere et al. (2018)*, it was also shown that individuals with depressive symptoms have difficulty regulating emotions, resulting in lower emotional complexity. Therefore, from a psychological perspective, information about emotions is useful in diagnosing mental disorders.

Unlike other approaches to mental health detection, such as DAS (Depression, Anxiety, Stress) detection models, EA models can provide greater consistency and enable the visualization of mood changes through published text, thereby avoiding false positives. For example, a semi-supervised ML model, DASentimental, has been proposed in *Fatima et al. (2021)* to extract cues to depression, anxiety, and stress from written text. However, it is only capable of identifying negative emotions.

Currently, several works on EA focus exclusively on the use of Paul Ekman's basic emotions. However, the importance of mental health at this time and the states associated with it, such as depression, loneliness, and hopelessness, are indicators that would help detect mental disorders. For this reason, in this article we compiled and evaluated a dataset labeled with 14 different emotions, offering a unique approach by including emotions and states that go beyond the basic emotions defined by Ekman (anger, disgust, fear, happiness, sadness, and surprise). Our approach to EA is based on creating a multi-classification dataset of 16 different emotions, covering emotions such as loneliness and hopelessness and then using transfer learning by fine-tuning different Spanish and multilingual LLMs. These models include (1) BETO (*Cañete et al., 2020*), (2) ALBETO (*Cañete et al., 2022*), (3) BERTIN (*de la Rosa et al., 2022*), (4) XLM (*Conneau et al., 2020*), (5) DistilBETO (*Cañete et al., 2022*), (6) MarIA (*Gutiérrez-Fandiño et al., 2022*), (7) multilingual BERT (*Devlin et al., 2019*), (8) multilingual DeBERTA (*He, Gao & Chen, 2021*), and (9) TwHIN (*Zhang et al., 2023b*). We also evaluate text generation models for text classification tasks, such as BLOOM (*Scao et al., 2022*), BART (*Lewis et al., 2020*), GPT-2 (*Radford et al., 2019*), and Llama-2 (*Touvron et al., 2023*).

## DATASET

In this research, a novel dataset is formulated by combining, translating, and relabeling pre-existing datasets using a detailed emotional classification system.

The first step in developing this corpus is to create a taxonomy of emotions. We start with Ekman's taxonomy, which includes negative emotions such as sadness and anger. Next, we include Plutchik's Wheel of Emotions, which is a broader representation than Ekman's and considers fine-grained emotions such as grief, disgust, or remorse. We complement this

taxonomy with research from *Leis et al. (2019)*, which identified a list of words that may indicate signs of depression. The selection of words was made by psychiatrists, members of the Institute of Neuropsychiatry and Addictions (INAD), Parc Salut Mar, Barcelona, Spain. It includes both the words and a global score assigned to them, which is the sum of the scores given by each evaluator on a Likert scale (from 1 to 5). The scores are based on the relevance of that word to a patient with depression. The alternative state or emotion that some of them could reflect was extracted. Some of the emotional states are depressed, disappointed, ashamed, hopeless, lonely, regretful, nervous, and suicidal. Keeping in mind that the terms provided could represent other conditions in addition to identifying signs of depression, the terms with the highest scores were extracted and included in the taxonomy. The final list of emotional states are: (1) angry, (2) depressed, (3) disappointed, (4) disgusted, (5) embarrassed, (6) fearful, (7) grieved, (8) hopeless, (9) joyful, (10) lonely, (11) nervous, (12) neutral, (13) regretful, (14) sad, (15) suicidal, and (16) surprised.

Once the emotions were identified, the next step was to find corpora that contained at least one or more of these emotions.

- Merging Datasets for EA (*De Arriba, Oriol & Franch, 2021*). The International Workshop on Software Engineering Automation: A Natural Language Perspective (NLP-SEA) presented this dataset in 2021. It contains 5,260 Spanish documents classified as:(1) sadness, (2) not relevant, (3) fear, (4) happiness, (5) anger, and (6) surprise. The dataset was collected from Twitter and it is related to the COVID-19 outbreak.
- Detecting signs of depression in tweets in Spanish: behavioral and linguistic analysis (*Leis et al., 2019*). It contains documents about detecting signs of depression in Spanish.
- Detecting Depression in Social Media Via Twitter Usage (https://github.com/ram574/Detecting-Depression-using-Tweets). This dataset is compiled from X and contains texts related to the labels depressed, hopeless, lonely, and suicidal. The keywords used to retrieve the dataset are depressed, depression, hopeless, lonely, suicide, and antidepressant.
- GoEmotions (*Demszky et al., 2020*). This dataset was collected by Google from popular English subreddits and manually tagged with 27 emotion categories, including 12 positive emotion categories, 11 negative, four ambiguous, and one neutral. To validate that the taxonomic choices were consistent with the given emotions, a principal component analysis was performed, which allowed two datasets to be compared by extracting linear combinations with greater variability.

Next, we show some translated examples from the dataset. There are several messages about coronavirus: *I think no one is going to forget the "COVID-19" pandemic... not even the children.* (sadness), *I'm so scared to go to my doctor's appointment tomorrow because I don't want to catch the coronavirus, I hope no one there has it when I go tomorrow.* (fear), or *The European Medicines Agency will make a statement shortly. Yes, there is a link between the ASTRA-ZENECA vaccine and blood clots. Better late than never.* (anger). Other examples are *I have been going to a Mexican restaurant in my neighborhood every month for the past 2.5 years, but no one knows my name.* (disappointment) and *Every night I cry in my*

**Table 1  Dataset statistics, including the number of emotions per label and split.**

| Emotional state | Train | Val | Test | Total |
|---|---|---|---|---|
| Angry | 775 | 259 | 259 | 1,293 |
| Depressed | 3,520 | 1,174 | 1,174 | 5,868 |
| Disappointed | 2,823 | 941 | 941 | 4,705 |
| Disgusted | 1,295 | 432 | 432 | 2,159 |
| Embarrassed | 730 | 243 | 244 | 1,217 |
| Fearful | 1,280 | 427 | 427 | 2,134 |
| Grieved | 184 | 61 | 62 | 307 |
| Hopeless | 739 | 247 | 247 | 1,233 |
| Joyful | 111 | 37 | 38 | 186 |
| Lonely | 2,931 | 977 | 977 | 4,885 |
| Nervous | 358 | 120 | 120 | 598 |
| Neutral | 3,358 | 1120 | 1,120 | 5,598 |
| Regretful | 596 | 199 | 199 | 994 |
| Sad | 639 | 213 | 214 | 1,066 |
| Suicidal | 3,786 | 1262 | 1,263 | 6,311 |
| Surprised | 3 | 1 | 1 | 5 |
| Total | 23,128 | 7,713 | 7,718 | 38,559 |

*bed and think of ways to kill myself. I have wanted to end my life for years, but I couldn't. If I didn't know that suicide is a sin, I would have done it long ago.* (suicidal).

The English sentences are translated into Spanish using SYSTRAN (https://www.systran.net/en/translate/). A manual check was then performed to verify their accuracy.

Another problem we faced was that the sentences from Reddit were too long for some models, such as Transformers. Since these documents were too complex, we decided to split them into sentences and annotate each sentence individually. The annotation phase was carried out by members of our research group, where each document was revised three times and the emotion of the sentence was decided in a final meeting.

The final step is a cleanup process that removes retweets, removes line breaks, mentions, hyperlinks, images, and ensures there are no duplicate documents.

Next, we split the dataset into training, validation, and test in a ratio of 60-20-20. Table 1 shows the statistics of the final dataset and the number of instances in each split. This dataset contains 38,559 sentences annotated with 16 emotions. It can be observed that the emotions are not balanced, and there is a significant lack of emotions labeled as surprise. Joy is another underrepresented emotion, with only 186 instances. However, sentences related to negative emotions, such as disappointment, lonely, and sadness contain several examples.

Figure 1 illustrates the information gain of the linguistic features extracted with UMUTextStats (*García-Díaz et al., 2022*) for each emotion[1]. It can be observed that the most informative features are related to stylometry, specifically the length of the corpus and the number of syllables and words. Other significant features are health-related lexical

[1] Please, refer to the 'Baseline' section for a description of the linguistic categories of this tool.

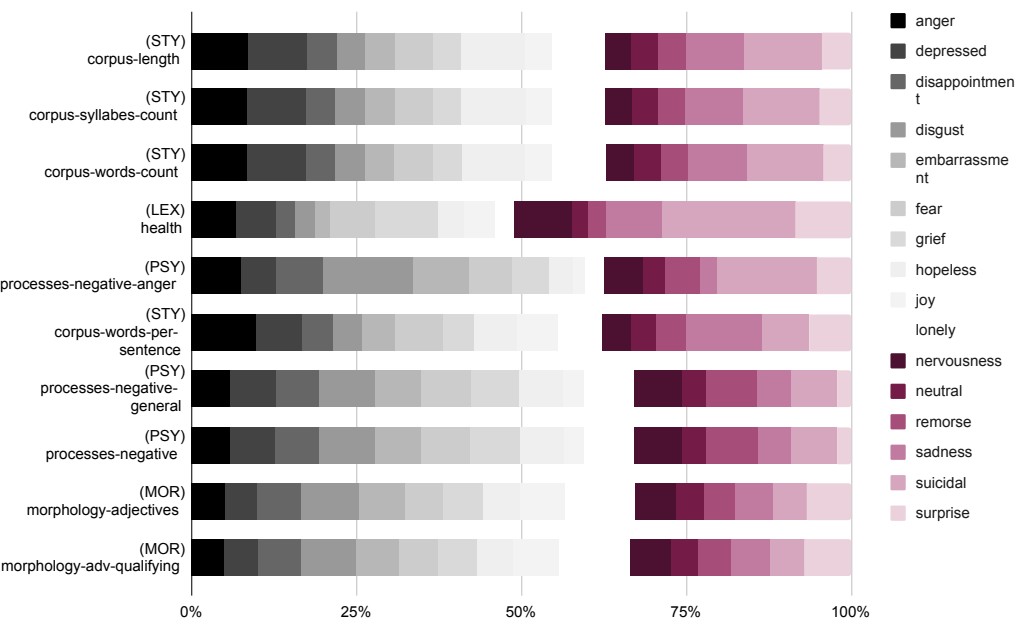

**Figure 1  Information gain of the top 10 linguistic features of the dataset.** The values are normalized in order to fit the 100%.

items, which are correlated with sadness and suicidal feelings. Other relevant features include qualifying adjectives and negative processes.

The compiled dataset is available to the scientific community (https://github.com/NLP-UMUTeam/peerj-fine-grain-emotion-analysis).

# MATERIALS AND METHODS

To evaluate the dataset, we mainly used two techniques that involve fine-tune different LLMs to determine the emotion. Specifically, we analyzed two different language models for the classification task: (1) encoder-only models based on masked language models (MLM), and (2) encoder–decoder autoregressive transformers for text generation, including BART, T5, GPT-2, BLOOM, and Llama-2. We have also tested several model combination techniques, including knowledge integration and ensemble learning.

Our pipeline is shown in Fig. 2. First, the data preprocessing module is used to cleanse the texts. Second, the splitter module separates the corpus into training, validation, and test sets. Third, feature sets are extracted to train the classification model. Fourth, we fine-tune different LLMs for emotion classification. Finally, we evaluate different ensemble learning methods for text classification.

## Data pre-processing and splitter

We used a variety of feature sets, including linguistic features (LFs) for the baseline and different types of embedding. A standard preprocessing procedure was performed, which

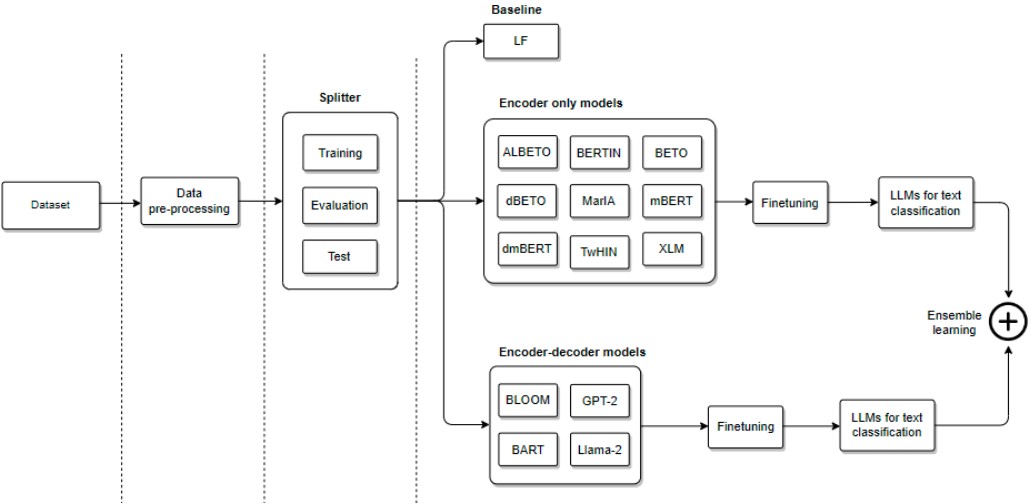

**Figure 2** General architecture of our pipeline for emotion classification.

consisted of removing all social media jargon, such as hyperlinks, hashtags, and mentions. In addition, all percentages and numbers were replaced with fixed tokens to prevent the classifiers from learning specific quantities. Finally, the normalized version is used to extract tokens of the embedding-based features and then to fine-tune various LLMs.

The splitter module plays a crucial role in extracting the training, development, and test datasets, and these splits vary depending on the task at hand. In this particular scenario, we have divided the dataset in a 60-20-20 ratio with stratified sampling to ensure that each emotion is correctly represented in each sample.

## Baseline

To compare the results, we created a baseline based on LFs. LFs are a means of representing documents as a vector formed by the percentages and raw counts of linguistically relevant features that indicate what a text says and how it says so (*Tausczik & Pennebaker, 2010*). To extract the LFs, we rely on UMUTextStats (*García-Díaz et al., 2022*). This tool captures 365 linguistic features, organized as follows:

- **Correction and style of written communication (COR)**. The analysis detects a variety of errors, including orthographic errors such as misspelled words, style issues, and performance errors. These performance errors can consist of sentences beginning with numbers or the same word, as well as identifying common errors and unnecessary phrases.
- **Phonetics (PHO)**. It captures expressive lengthening, in which certain letters are deliberately extended to emphasize their meaning.
- **Morphosyntax (MOR)**. The structure of words is recorded along with grammatical attributes, including gender and number, and various affixes, such as nominal, adjectival, verbal, adverbial, augmentative, diminutive, and derogatory suffixes. In addition, these
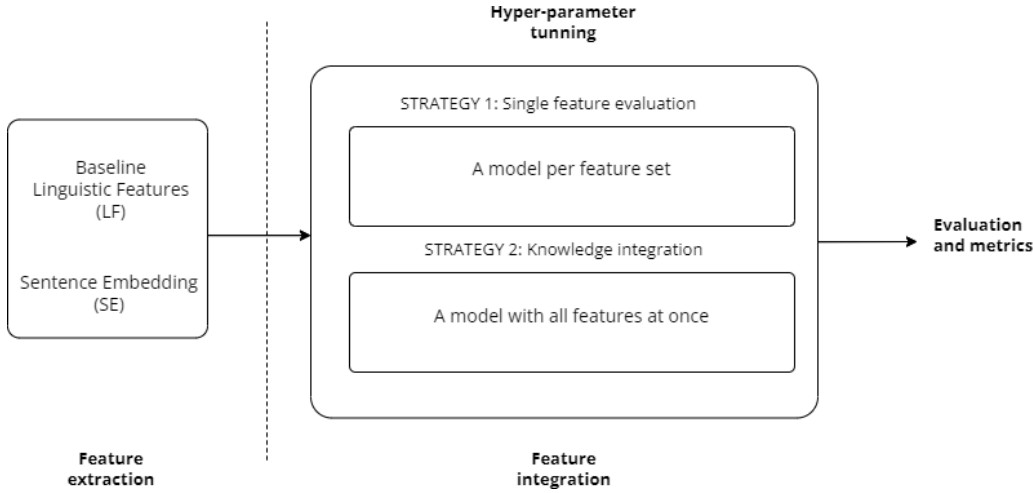

**Figure 3** General architecture of the feature-based classification model for EA.

features are organized according to their respective part-of-speech categories, such as verbs, nouns, and adjectives.

- **Semantics (SEM)**. Includes sound words, polite expressions, derogatory terms, and figures of speech in which the part represents the whole.
- **Pragmatics (PRA)**. The use of figurative language devices, including understatement, rhetorical questions, hyperbole, idiomatic expressions, verbal irony, metaphors, and similes.
- **Stylometry (STY)**. It records punctuation symbols, corpus statistics, and metrics related to the number of words, syllables, or sentences.
- **Lexical (LEC)**. The text provides a thorough identification and analysis of the topics, covering both abstract and general topics.
- **Psycho-linguistic processes (PLI)**. This category contains emoticons and lexicons related to emotions and feelings.
- **Register (REG)**. It emphasizes the distinction between informal and formal language, and addresses topics that may be considered offensive.
- **Social media jargon (SOC)**. This category captures features related to the speaker's mastery of social media jargon.

For the training the baseline based on the LFs, we relied on a multilayer perceptron (MLP) because these features do not contain sequential information such as text.

### Encoder-only classification model

We evaluated several encoder-only language models. Figure 3 shows the pipeline of our approach as well as different feature-integration strategies, such as knowledge integration. Regarding sentence embeddings, several pre-trained transformer-based models are evaluated, which can be broadly classified into BERT-based and RoBERTa-based models. The main difference between these two architectures is that RoBERTa performs masking during training, whereas BERT performs masking at the beginning of training.

Transformer-based models can be pre-trained with a multilingual corpus or with a monolingual corpus of a specific language. Thus, we have performed an evaluation of several multilingual and monolingual Spanish models for this task. It is possible to categorize models using BERT and RoBERTa-based architectures as follows:

- **BERT-based monolingual model**. The study used the Spanish variant of BERT called BETO (*Cañete et al., 2020*). The evaluation of BETO model also included two lighter versions derived from it: ALBETO and DistilBETO (*Cañete et al., 2022*).
- **BERT-based multilingual model**. In terms of BERT-based models pre-trained with a multilingual corpus, the models used are: (1) TwHIN-BERT (*Zhang et al., 2023b*), a multilingual tweet language model trained with 7 billion tweets from more than 100 different languages; (2) M-BERT (*Devlin et al., 2019*), a transformer model pre-trained with a large corpus of multilingual data in a self-supervised manner; (3) M-DistilBERT (*Sanh et al., 2019*), a distilled version of the multilingual BERT base model.
- **RoBERTa-based monolingual model**. In this article, we have evaluated different models pre-trained with the Spanish corpora, such as MarIA (*Gutiérrez-Fandiño et al., 2022*) and BERTIN (*de la Rosa et al., 2022*).
- **RoBERTa-based multilingual model**. Regarding RoBERTa-based models pre-trained with a multilingual corpus, we evaluated XLM-RoBERTa (100–1,280) (*Lample & Conneau, 2019*), which is a multilingual version of RoBERTa trained with data filtered from CommonCrawl from 100 different languages.

We fine-tuned each model separately for each feature set using hyperparameter tuning. To do this, we use a network architecture design in which each feature set is connected to a separate stack of hidden layers. The output of each layer is then concatenated and connected to a new set of hidden layers, which in turn are connected to the final output layer. We select the best model based on the validation split.

In order to produce more robust solutions, we evaluate one feature integration strategy known as knowledge integration. This strategy involves training a multi-input neural network from scratch, incorporating all the sentence embeddings for each encoder-only mode at once. The idea behind this approach is that the network learns during training how to exploit the strengths of each feature set.

## Encoder–decoder models

LLMs have revolutionized many aspects of NLP. LLMs possess few-shot and even zero-shot learning capabilities *Brown et al. (2020)*. New models such as Llama-2 (*Touvron et al., 2023*) and GPT-3 (*Brown et al., 2020*), which have been trained on large and diverse text corpora, have further enhanced these capabilities. This has opened up a wide range of possibilities in the field of NLP, such as direct label prediction through prompting. In fact, several articles have tested this hypothesis and achieved good performance across a wide variety of NLP tasks (*Plaza-del Arco, Nozza & Hovy, 2023*; *Wang et al., 2023*).

In this study, we evaluate several generative LLMs such as GPT-2 (*Radford et al., 2019*), BLOOM (*Scao et al., 2022*), BART (*Lewis et al., 2020*), and Llama-2 (*Touvron et al., 2023*). These models were pre-trained on massive amounts of text data, enabling them to generate

coherent and contextually relevant text in ER tasks. However, since we have a total of 16 possible emotions and some of them are similar, we apply a fine-tuning approach instead of a contextual learning approach such as ZSL or FSL. Fine-tuning involves taking a pre-trained LLM and adapting it to a specific task or domain. This is done by training the LLM on a smaller dataset that is specific to the task or domain and adjusting the model weights and parameters to better fit the new data. In this way, the model can achieve good performance without the need for extensive training data or abundant computational resources. The models tested in this study are the following:

- **BLOOM**. It is an autoregressive LLM that has been trained on extensive text data using industrial-scale computing resources. It excels at generating coherent text in 46 languages and 13 programming languages, making it nearly indistinguishable from human-generated text (*Scao et al., 2022*). In this article, we have used a smaller version of BLOOM, known as BLOOM-3B (https://huggingface.co/bigscience/bloom-3b), which has 3 billion parameters.
- **GPT-2**. It is a pre-trained model using a causal language modeling (CLM) target and a Transformer-based model that has undergone extensive pre-training on large corpora in a self-supervised manner (*Radford et al., 2019*). In this case, we used a Spanish version of GPT-2 called "Spanish GPT-2 (https://huggingface.co/mrm8488/spanish-gpt2)". This model was trained from scratch on the large Spanish corpus, also known as the BETO corpus, using Flax.
- **BART**. It is a transformer encoder–decoder (seq2seq) model with a bidirectional encoder, similar to BERT, and an autoregressive (GPT-like) decoder. BART is pre-trained by two key steps: (1) introducing noise into the text using a flexible noise function, and (2) training a model to recover the original, uncorrupted text. This model shows remarkable effectiveness when fine-tuned for text generation tasks such as summarization and translation. However, it also performs well in comprehension tasks, including text classification and question-answering. Specifically, we have used *mBART-50*, which is a pre-trained multilingual Sequence-to-Sequence model using the "Multilingual Denoising Pretraining" target (*Lewis et al., 2020*).
- **Llama-2**. Llama 2 contains a set of pre-trained and fine-tuned generative text models with parameters ranging from 7 to 70 billions. In this article we have used a version of 7B optimized for the dialog cases (*meta-llama/Llama-2-7b-chat-hf*) (*Touvron et al., 2023*).

## Ensemble learning

After acquiring different models for identifying emotions, we evaluated different techniques for ensemble learning, which consists of combining predictions from many individual estimators to produce a more robust estimator. During our research, we investigated two methods of averaging to combine predictions from the best encoder-only model and the best encoder–decoder model by (1) computing the mean, which averages the probabilities produced by each model; or (2) selecting the label with the highest probability, which involves observing the probabilities associated with each model and selecting the one with the highest probability.

# RESULTS AND ANALYSIS

In this section, we present and discuss the results obtained using the encoder-only and encoder–decoder strategies and the feature integration techniques with the test split and compared to the baseline.

Since our problem involves unbalanced classification, we evaluate the performance of the classification models using both the macro average F1 score and the weighted average F1 score. These metrics serve as indicators of the performance of a classification model in terms of accuracy and recall, albeit with different computational methods and weighting schemes. The macro average F1 score evaluates both the precision and recall of each class, combining the results equally without regard to class imbalance. In this way, it allows for the selection of the best model that performs equally well for all labels. The weighted average F1 score is a widely used metric in classification problems, especially when dealing with unbalanced datasets where some classes may have many more examples than others. Unlike the macro average F1 score, this metric accounts for class imbalance by assigning a different weight to each class based on its frequency in the dataset. Therefore, in our work, we consider the macro average F1 score as the metric to determine the best model, but we keep the weighted average F1 score to reflect the overall performance of the model.

## Results of encoder-only models

First, we report the results of several fine-tuned encoder-type models mentioned in 'Encoder-only classification model'. For each model, we added a dense sequential classification layer with the same number of neurons as the output classes for fine tuning and performed hyperparameter optimization to find the best training parameters for these models using the validation split. The hyperparameters under consideration, along with their respective interval ranges, are: (1) weight decay (ranging from 0 to 0.3); (2) training lot size (ranging from 8 to 16); (3) number of training epochs (ranging from 1 to 6); and (4) learning rate (ranging from 1e−5 to 5e−5).

Table 2 shows the best set of hyperparameters obtained for each encoder-only model. It can be observed that most of the models, including ALBERT, BERTIN, Distilled mBERT, TwHIN-BERT, and XLM-RoBERTa, performed better with a training batch size of 16 and a lower learning rate. Most of these models also performed better with a warm-up step of 500, with the exception of BERTIN, BETO, and TwHIN-BERT, which performed better with a value of 250, and Distilled mBERT and XLM-RoBERTa, which performed best with a warm-up step of 0. In terms of weight decay, most of the models performed better with a value lower than 0.2.

Table 3 shows the results obtained by the encoder-only models based on Transformers and the baseline. As expected, all models outperformed the baseline in terms of M-F1. In addition, the RoBERTa architecture consistently outperformed BERT. The top two scores were achieved by MarIA (60.47% in M-F1 and 79.42% in W-F1) and the knowledge integration model (58.70% in M-F1 and 78.91% in W-F1). In addition, XLM-RoBERTa outperforms the multilingual BERT (58.27% vs. 56.23% in M-F1). Notably, the lightweight versions of BETO, ALBETO and Distilled mBERT yielded limited results.

**Table 2** Best subset of hyperparameters for each encoder-only model based on Transformers.

|  | Learning rate | Epoch | Batch size | Warmup steps | Weight decay |
|---|---|---|---|---|---|
| ALBETO | 3.5e−05 | 5 | 16 | 500 | 0.0006 |
| BERTIN | 2e−05 | 5 | 16 | 250 | 0.19 |
| BETO | 2.6e−05 | 4 | 8 | 250 | 0.00089 |
| Distilled BETO | 3.8e−05 | 3 | 8 | 500 | 0.21 |
| MarIA | 1.4e−05 | 4 | 8 | 500 | 0.069 |
| mBERT | 2.1e−05 | 3 | 8 | 500 | 0.25 |
| Distilled mBERT | 3.6e−05 | 5 | 16 | 0 | 0.26 |
| TwHIN-BERT | 2.7e−05 | 4 | 16 | 250 | 0.00076 |
| XLM | 3.7e−05 | 2 | 16 | 0 | 0.098 |

**Table 3** Benchmark of the different pre-trained models and linguistic feature-based model. The metrics reported for each model and dataset include macro precision (M-P), macro recall (M-R), macro F1 score (M-F1), and weighted F1 score (W-F1).

|  | Architecture | Language | M-P | M-R | W-F1 | M-F1 |
|---|---|---|---|---|---|---|
| LF (baseline) | – | Mono | 37.6551 | 41.8719 | 56.2573 | 36.9939 |
| ALBETO | BERT | Mono | 56.2657 | 56.3588 | 76.5421 | 55.7226 |
| BERTIN | RoBERTa | Mono | 56.6446 | 56.6979 | 76.9988 | 55.9180 |
| BETO | RoBERTa | Mono | 58.3675 | 57.8535 | 78.2100 | 58.0378 |
| Distilled BETO | BETO | Mono | 57.2375 | 58.0291 | 77.1875 | 57.4865 |
| MarIA | RoBERTa | Mono | **61.9512** | **59.4110** | **79.4245** | **60.4771** |
| mBERT | RoBERTa | Multi | 56.5545 | 56.9628 | 75.8372 | 56.2317 |
| Distilled mBERT | BERT | Multi | 53.7665 | 56.3917 | 74.1021 | 54.0067 |
| TwHIN-BERT | BERT | Multi | 59.0658 | 57.8394 | 78.7239 | 58.2436 |
| XLM | RoBERTa | Multi | 56.6756 | 58.2797 | 77.5739 | 58.2797 |
| KI | – | – | 59.1886 | 59.3930 | 78.9181 | 58.7096 |

**Notes.**
Best results are in bold.

When comparing the results of transformers trained on monolingual datasets with those trained on multilingual datasets, a slight advantage is observed for the models trained only in Spanish. This suggests that obtaining specific pre-trained models for the target language is preferable to using multilingual variants.

## Results of encoder–decoder models

In this study, we also evaluated different encoder–decoder models for text generation in EA tasks due to the promising results reported in several studies (*Plaza-del Arco, Nozza & Hovy, 2023*; *Brown et al., 2020*). Since these are Sequence-to-Sequence (Seq2Seq) models, *i.e.,* they take input text and produce output text, we added a linear layer to the pooled output for fine-tuning to ensure that the output corresponds to one of the emotions.

Some of these text generation models, such as BLOOM and Llama-2, are multilingual and pre-trained on a large corpus. The size of these models is quite large. BLOOM-3b has three billion parameters and weighs about 6 GB, while Llama-2 has 7 billion parameters and weighs about 13.5 GB. Therefore, we used the LoRA (*Hu et al., 2021*) approach to

**Table 4  Benchmark of the different encoder–decoder models.** Metrics reported for each model and dataset include macro precision (M-P), macro recall (M-R), macro F1 score (M-F1), and weighted F1 score (W-F1).

|  | M-P | M-R | W-F1 | M-F1 |
|---|---|---|---|---|
| LF (baseline) | 37.6551 | 41.8719 | 56.2573 | 36.9939 |
| Spanish GPT-2 | **57.1349** | **57.1106** | **77.2334** | **57.0592** |
| BLOOM-3b | 49.2585 | 45.9486 | 65.1833 | 45.5222 |
| Llama-2 | 52.0913 | 52.5977 | 73.8725 | 51.2868 |
| mBART | 52.4731 | 52.1906 | 73.8514 | 52.1617 |

**Notes.**
Best results are in bold.

speed up the fine-tuning process and reduce memory consumption. LoRA is based on representing weight updates with two smaller matrices called *update matrices* by low-rank decomposition. These new matrices can be trained to adapt to new data while keeping the total number of changes small. The original weight matrix remains frozen and does not receive any further adjustments. To produce the final results, both the original and the adjusted weights are combined. Due to the size of the encoder–decoder models, we only performed hyperparameter optimization with the validation split for the epoch parameter within a range of 5 epochs. We kept other parameters constant: 0.01 for weight decay, $2e-5$ for learning rate, and 500 warm-up steps.

Table 4 shows the results obtained with different encoder–decoder models. As can be seen, the best results are obtained with the Spanish GPT-2 (57.06% in M-F1 and 77.23% in W-F1). In addition, the Spanish GPT-2 is the only monolingual model and the lightest among the models compared. Therefore, we can draw the same conclusion as in the previous case (see 'Results of encoder-only models'): obtaining specific pre-trained models for the target language is preferable to using multilingual variants. Regarding the multilingual models, the best result was obtained with multilingual BART, with an M-F1 of 52.16%.

## Results of ensemble learning

In this section, different ensemble learning techniques are evaluated using the best encoder-type model (MarIA) and encoder–decoder-type model (Spanish GPT-2) for EA. Table 5 shows the results obtained. As can be seen, both the mean-based and the highest probability-based techniques have improved the weighted F1 score compared to MarIA, with an improvement of 0.0556% and 0.0227%, respectively. For the Spanish GPT-2, the ensemble learning techniques improved the weighted F1 score, with improvements of 2.25% for the mean-based approach and 2.21% for the highest likelihood-based approach. However, ensemble learning did not improve the macro F1 score metrics because it did not improve predictions in emotion classes with fewer instances in the test set, such as *joy* and *grief*. In contrast, it did improve predictions in some cases, such as the *depressed* emotional state, which has more instances in the test set.

Table 6 shows the classification reports for the best single model, MarIA, and the best ensemble-learning model, mean-based ensemble learning. The results show that both

**Table 5** **Benchmark of the different ensemble learning techniques between the best pre-trained (MarIA) and encoder–decoder (Spanish GPT-2) model.** The reported metrics for each model and dataset include macro precision (M-P), macro recall (M-R), macro F1 score (M-F1), and weighted F1 score (W-F1).

| | M-P | M-R | W-F1 | M-F1 |
|---|---|---|---|---|
| LF (baseline) | 37.6551 | 41.8719 | 56.2573 | 36.9939 |
| MarIA | **61.9512** | **59.4110** | 79.4245 | **60.4771** |
| Spanish GPT-2 | 57.1349 | 57.1106 | 77.2334 | 57.0592 |
| Mean | 61.5379 | 58.9295 | **79.4801** | 60.0464 |
| Highest probability | 61.5675 | 58.8974 | 79.4472 | 60.0270 |

**Notes.**
Best results are in bold.

**Table 6** **Classification report of Precision (P), Recall (R), and F1 score (F1) of MarIA (left) and Ensemble based on the mean (right) with the test set for each emotion.**

| Emotion | P | R | F1 | P | R | F1 |
|---|---|---|---|---|---|---|
| | | MarIA | | | Ensemble (mean) | |
| Angry | 77.5934 | 72.2008 | 74.8000 | 77.6860 | 72.5869 | 75.0499 |
| Depressed | 93.1330 | 92.4190 | 92.7746 | 93.4708 | 92.6746 | 93.0710 |
| Disappointed | 54.8295 | 61.5303 | 57.9869 | 56.9573 | 65.2497 | 60.8222 |
| Disgusted | 44.2516 | 47.2222 | 45.6887 | 45.4756 | 45.3704 | 45.4229 |
| Embarrassed | 34.2466 | 30.7377 | 32.3974 | 35.9813 | 31.5574 | 33.6245 |
| Fearful | 60.8911 | 57.6112 | 59.2058 | 60.6061 | 56.2061 | 58.3232 |
| Grieved | 41.0256 | 25.8064 | 31.6832 | 38.7755 | 30.6452 | 34.2342 |
| Hopeless | 86.2222 | 78.5425 | 82.2034 | 87.9630 | 76.9231 | 82.0734 |
| Joyful | 45.4545 | 39.4737 | 42.2535 | 38.4615 | 26.3158 | 31.2500 |
| Lonely | 96.0685 | 97.5435 | 96.8004 | 95.6871 | 97.6459 | 96.6565 |
| Nervous | 21.9780 | 16.6667 | 18.9573 | 22.2222 | 18.3333 | 20.0913 |
| Neutral | 93.6057 | 94.1071 | 93.8557 | 90.8457 | 93.0357 | 91.9277 |
| Regretful | 64.9425 | 56.7839 | 60.5898 | 62.3596 | 55.7789 | 58.8859 |
| Sad | 80.9091 | 83.1776 | 82.0276 | 81.4480 | 84.1121 | 82.7586 |
| Suicidal | 96.0692 | 96.7538 | 96.4102 | 96.6667 | 96.4371 | 96.5517 |
| Surprised | 0.0 | 0.0 | 0.0 | 0.0 | 0.0 | 0.0 |

strategies are similar for fine-grained EA. The big difference is observed in the *joy* emotion, where MarIA achieves an F1 score of 42.2535%, while the mean-based ensemble learning approach achieves only 31.250%. In terms of precision and recall, both strategies show similar performance. However, ensemble learning based on the mean shows a larger difference between precision and recall for the identification of texts labeled as *joy*. Also, neither strategy was able to identify the only instance of *surprise* in the test split.

Regarding the identification of negative emotional states such as *depressed*, *disappointed*, *embarrassed*, *grieved*, *nervous*, *sad*, and *suicidal*, we can see that the MarIA model performs well in identifying the emotional states most directly related to mental disorders, such as *depressed*, *sad*, and *suicidal*, exceeding 80% accuracy.

Next, we analyze the limitations of these models in predicting emotions using the confusion matrix to evaluate the misclassified examples. First, regarding the MarIA model (see Fig. 4), it can be observed that MarIA performs very well in predicting certain emotional states such as *depressed*, *lonely*, *neutral*, and *suicidal*, with an accuracy rate of over 90%. However, for more confusing and difficult-to-identify emotions, such as *embarrassment*, *grief*, *joy*, *nervousness*, and *surprise*, the model tends to confuse them with other similar emotions, achieving less than 40% accuracy for these emotions. In most cases, MarIA confuses these emotions with the emotion *disappointment*. Secondly, regarding the Spanish GPT-2 (see Fig. 5), it can be observed that its performance is similar to that of MarIA. The Spanish GPT-2 tends to make the same classification errors, except in the case of *surprise*, where the model tends to confuse *surprise* with *embarrassment*. Third, regarding the mean-based ensemble learning model (see Fig. 6), it can be observed that ensemble learning has improved the predictions for the emotional states *angry*, *depressed*, *disappointed*, *disgusted*, *nervous*, and *sad* by 1–3%, while it has worsened the predictions for the other emotions in the same range. The only exception is *grief*, which is 5% worse off. For this reason, the weighted F1 score of the ensemble learning is higher than that of MarIA, but not the macro F1 score. Moreover, we can see that the ensemble learning has improved for the same emotions that MarIA has improved over the Spanish GPT-2 model, but to a greater extent (between 1% and 8%).

By analyzing the results and errors, we can see that pre-trained encoder models have achieved better performance than encoder–decoder models for text generation. Furthermore, it has been shown that the knowledge integration strategy improves the performance of most models, but it does not manage to improve MarIA separately. Regarding the ensemble learning techniques, we can observe that combining the outputs by taking the mean of MarIA and Spanish GPT-2 improves the performance of Spanish GPT-2, both in terms of weighted and macro F1 scores. It has also improved the performance of predicting some of the more common emotions in the test set for MarIA, resulting in an improvement in the weighted F1 score metric but not in the macro F1 score. Thus, using the macro average F1 score as the reference metric, the MarIA model has achieved the best result at 60.48%.

Finally, Table 7 shows examples of some common misclassifications made by the MarIA model. It is noticeable that the model often confuses emotions such as *embarrassment*, *grief*, and *nervousness* with the emotion of *disappointment*, as shown in Fig. 4. This is because distinguishing between these emotions based on text alone is a challenging task, as shown in Table 7. Also, the emotion of disappointment entails more examples in the training set as compared to other emotions.

## Evaluation with the EmoEvalES 2021 dataset

To measure the robustness of our pipeline, we have evaluated the performance of both strategies (encoder-only and encoder–decoder) with the EmoEvalEs 2021 dataset. Table 8 shows the results obtained. According to *Plaza-Del-Arco et al. (2021)*, the reference metric used for the evaluation is accuracy. Our best results indicate that the fine-tuning approach of MarIA and BLOOM-3b have surpassed the best previous result for this task (GSI-UPM with

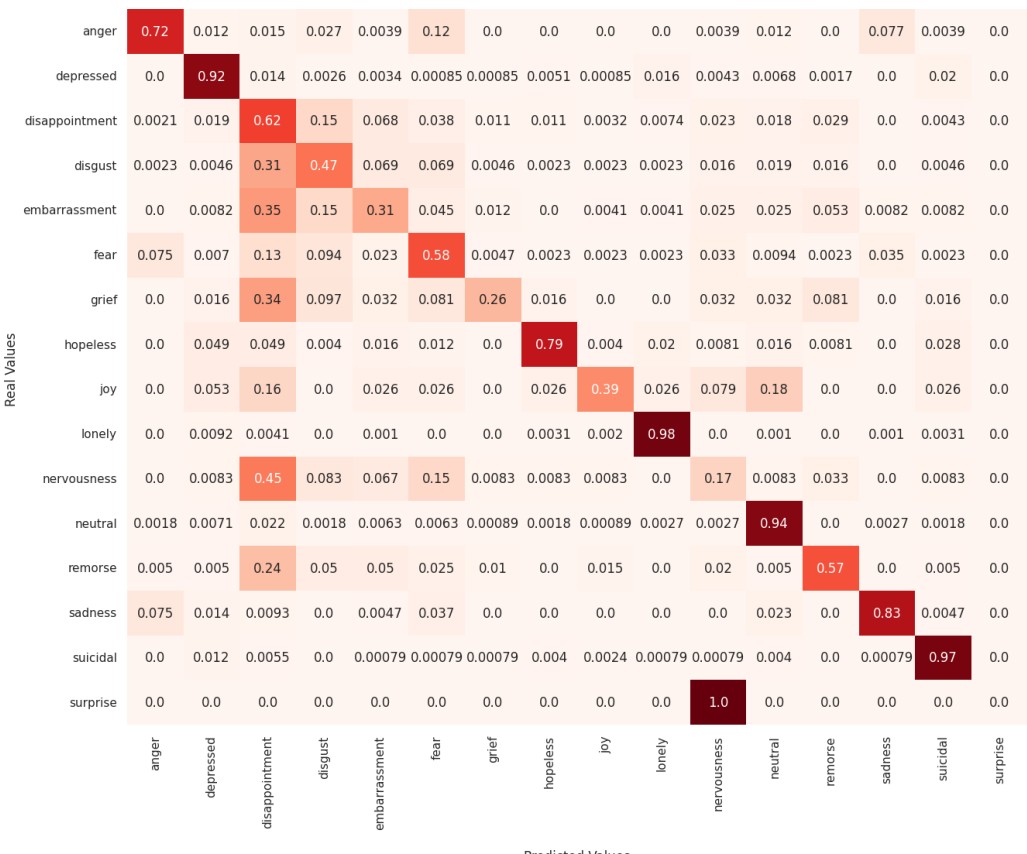

**Figure 4** Confusion matrix of the MarIA model.

an accuracy of 72.77%), achieving an accuracy of 73.5507% and 73.1280%, respectively. It draws our attention that in the EmoEvalES 2021 shared task, the GSI-UPM team achieved their best result by fine-tuning XLM-RoBERTa, but our results with this architecture are very limited (both in test and in custom validation). After reviewing the working notes of the GSI-UPM team, we learned that they used a version of XLM, but pre-trained with millions of tweets (*Barbieri, Espinosa Anke & Camacho-Collados, 2022*). It is possible that the limited results are related to the complexity of the model, as it contains 16 layers, 16 attention heads, and 1,280 hidden states, but further research should be conducted to identify the real cause of the limited results of this model with the EmoEvalES 2021 dataset.

After this analysis, we obtained the following results regarding the posed RQs. Regarding RQ1 about measuring the reliability of identifying negative fine-grained emotions, the results indicate that both precision and recall are usually good for some negative emotions (all the emotional states in the dataset except for *neutral*, *joyful* and *surprised*). However, the developed EA models often fail to classify documents labeled as *disappointment*, confusing these emotions with others such as *disgust*, *embarrassment*, our *nervousness*. Regarding RQ2, to determine the best approach to face EA using the text, the best macro average F1 score is achieved with the fine-tuning of MarIA, a monolingual Spanish LLM based on

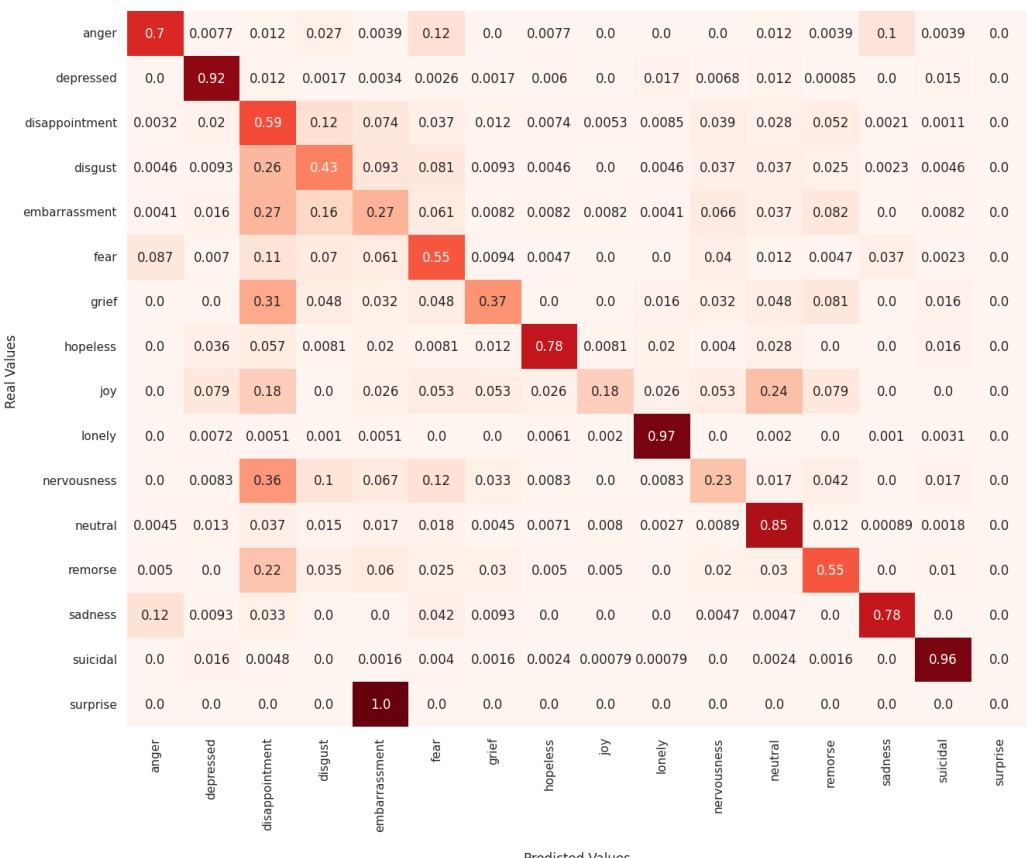

**Figure 5** Confusion matrix of the Spanish GPT-2 model.

the RoBERTa architecture. This strategy slightly outperformed other approaches based on fine-tuning generative language models. However, the ensemble learning strategy showed better results than MarIA in the negative emotional states: (1) *depressed*, (2) *disappointed*, (3) *embarrassed*, (4) *grieved*, (5) *nervous*, (6) *sad*, and (7) *suicidal*. In this sense, we also posed RQ3, which asked whether generative models are effective for EA, since they showed similar performance as the fine-tuned models. However, the combination of the results outperforms the ones obtained by the individual models, as compared to our experiment where we combined all the fine-tuned LLMs using a knowledge integration strategy, in which we observed a degradation of performance compared to MarIA.

## CONCLUSIONS AND FURTHER WORK

Mental health issues are a major global public health concern. Social media platforms are often one of the means by which users or individuals express their difficulties and find emotional support. Previous computational studies have consistently shown that individuals with mental disorders exhibit changes in their language and behavior, including a higher prevalence of negative emotions and a more intense self-focus. Therefore, clues

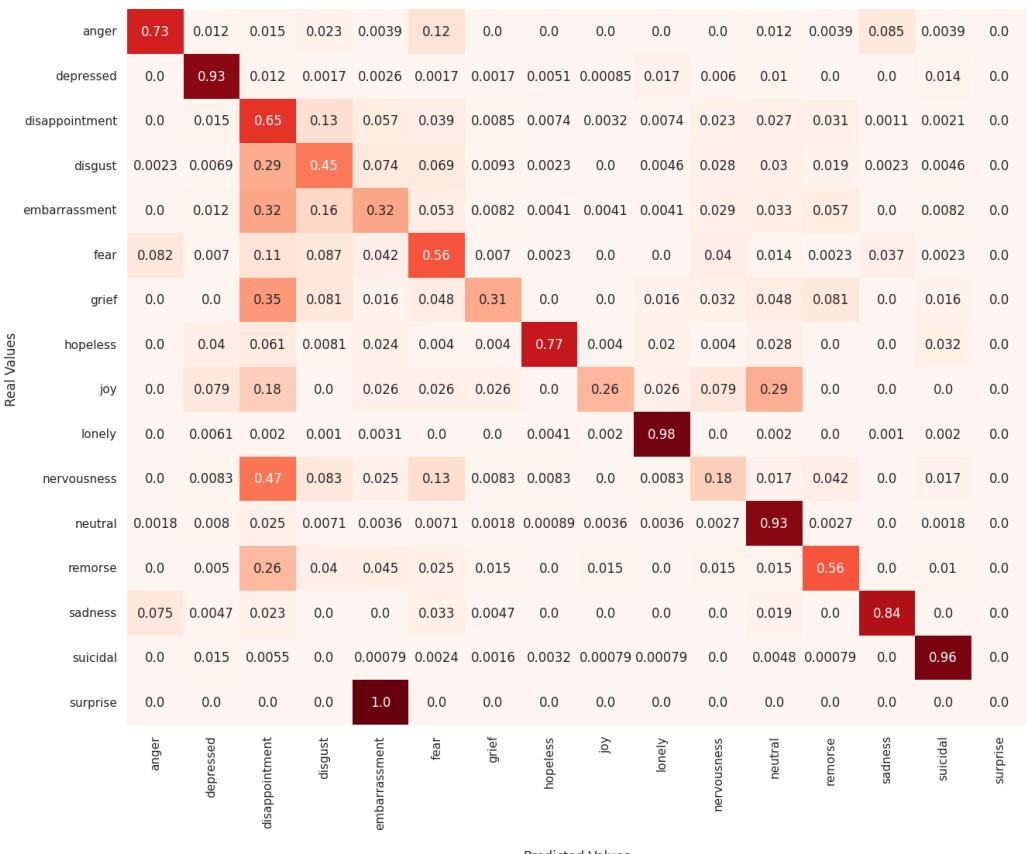

**Figure 6  Confusion matrix of the ensemble learning mean-based model.**

to an individual's altered mental state may be evident in their online posts, which often convey negativity.

This work presents two significant contributions to the field of EA for the detection of mental disorders in Spanish. To this end, we have created a novel corpus of 16 different emotions and performed an in-depth evaluation of several feature sets including linguistic features and transformer-based models based on encoders and encoder–decoder. We have also tested different techniques for feature integration, such as knowledge integration and ensemble learning, to see if they improve the performance of the models. Our results show that the fine-tuning approach of the encoder-only MarIA model has achieved the best result, with a macro F1 score of 60.48%.

As a limitation, statistical tests should be conducted to perform a better comparison of the models, since using a fixed validation split can bias some decisions about the best performing models. In this sense, we propose to extend this work using nested cross-validation for a better comparison. Moreover, as commented during the corpus compilation and annotation process, we split long documents because of the maximum length limitations of Transformers. In this sense, we will evaluate other strategies to handle long documents, such as pooling sentence embeddings or using Longformers.

**Table 7  Error analysis with some examples of misclassifications done by MarIA.** The original text and a literal translation into English using the Google Translate service are shown.

| # | Text | Truth | Prediction |
|---|------|-------|-----------|
| 1 | ¡Gracias por explicarlo! No estoy seguro de por qué mi comentario fue rechazado Sin contexto, no tiene sentido (*Thanks for explaining it! Not sure why my comment was downvoted Without context, it doesn't make sense.*) | embarrassment | disappointment |
| 2 | Maldita sea, realmente necesitas una novia, amigo (*Damn, you really need a girlfriend, dude*) | embarrassment | disappointment |
| 3 | Maldita sea, [NOMBRE] se dejó llevar (*Damn, [NAME] got carried away*) | grief | disappointment |
| 4 | Qué pobre mujer que solo quería decirle a alguien qué hacer (*What a poor woman who just wanted to tell someone what to do.*) | grief | disappointment |
| 5 | No tengo confianza para impulsar (*I don't have the confidence to push*) | nervousness | disappointment |
| 6 | Pero en serio, probablemente te vas a sentir muy mal (*But seriously, you're probably going to feel really bad.*) | nervousness | disappointment |
| 7 | Pero eventualmente ese mismo amigo más cercano apareció en mi puerta y se negó a irse (*But eventually that same closest friend showed up at my door and refused to leave.*) | joy | disappointment |

**Table 8  Benchmark of different encoder–decoder models for the EmoEvalEs dataset compared to the best approach in the official leaderboard.** Metrics reported for each model and dataset include accuracy.

| Strategy | Approach | Accuracy | % Difference |
|----------|----------|----------|-------------|
| GSI-UPM | XLM | 72.7657 | – |
| | ALBETO | 70.8333 | −1.9324 |
| | BERTIN | 68.4179 | −4.3478 |
| | BETO | 70.3502 | −2.4155 |
| | Distilled BETO | 70.5314 | −2.2343 |
| Encoder-only | **MarIA** | 73.5507 | +0.7850 |
| | mBERT | 66.4251 | −6.3406 |
| | Distilled mBERT | 64.1908 | −8.5749 |
| | TwHIN | 71.6787 | −1.0870 |
| | XLM | 18.5990 | −54.1667 |
| | Spanish GPT-2 | 69.6256 | −3.1401 |
| | **BLOOM-3b** | 73.1280 | +0.3623 |
| | Llama-2 | 70.1087 | −2.657 |
| | mBART | 64.4968 | −8.2689 |

**Notes.**
Best results are in bold.

As for future research, we plan to expand the dataset to include less common emotions such as *surprise*, *grief*, and *nervousness* in texts from different platforms, and to reduce the bias towards negative emotions. In addition, we plan to explore different data augmentation techniques and investigate how emotions are expressed within entities and their relationships in order to discover the emotions expressed by users.

### Funding

This work is part of the research projects LaTe4PoliticES (PID2022-138099OB-I00) funded by MICIU/AEI/10.13039/501100011033 and the European Fund for Regional Development (ERDF) -a way to make Europe and LTSWM (TED2021-131167B-I00) funded by MICIU/AEI/10.13039/501100011033 and by the European Union NextGenerationEU/PRTR. The funders had no role in study design, data collection and analysis, decision to publish, or preparation of the manuscript.

### Grant Disclosures

The following grant information was disclosed by the authors:
LaTe4PoliticES (PID2022-138099OB-I00): MICIU/AEI/10.13039/501100011033.
The European Fund for Regional Development (ERDF)-a way to make Europe and LTSWM (TED2021-131167B-I00): MICIU/AEI/10.13039/501100011033.
The European Union NextGenerationEU/PRTR.

### Competing Interests

The authors declare there are no competing interests.

### Author Contributions

- Alejandro Salmerón-Ríos conceived and designed the experiments, performed the experiments, performed the computation work, authored or reviewed drafts of the article, and approved the final draft.
- José Antonio García-Díaz performed the experiments, analyzed the data, prepared figures and/or tables, authored or reviewed drafts of the article, and approved the final draft.
- Ronghao Pan performed the experiments, analyzed the data, performed the computation work, prepared figures and/or tables, authored or reviewed drafts of the article, and approved the final draft.
- Rafael Valencia-García conceived and designed the experiments, analyzed the data, authored or reviewed drafts of the article, and approved the final draft.

### Data Availability

   The raw data and code is available in the Supplemental Files.

### Supplemental Information

Supplemental information for this article can be found online at http://dx.doi.org/10.7717/peerj-cs.1992#supplemental-information.

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
