# Peer review of "Fine grain emotion analysis in Spanish using linguistic features and transformers"

_PeerJ Computer Science, doi:10.7717/peerj-cs.1992_

## Round 0.1 · original submission · Major Revisions

The reviewers recommend significant improvements both in the technical rigour and in the clarity of the manuscript. Please follow all their recommendations closely.

**Language Note:** The review process has identified that the English language must be improved. PeerJ can provide language editing services - please contact us at copyediting@peerj.com for pricing (be sure to provide your manuscript number and title). Alternatively, you should make your own arrangements to improve the language quality and provide details in your response letter. – PeerJ Staff

·

Basic reporting

The article is written clearly in professional English but it needs to be checked for any typos and missing spaces between parentheses, words, and commas such as Line 154 and Line 104.

The article is easy to follow and the raw data were shared. The methods utilized and the datasets used are explained clearly but there are a couple of shortcomings:

The abstract lacks mention of NLP approaches, social media, and text modality. It would be clearer if these keywords were mentioned in the abstract as well.

Line 39: References are needed to support the claim that "there are remarkable results demonstrated with transformer-based models".

Line 92: Arousal refers to the strength or the intensity of the emotion and excitement is an emotion. The authors used "level of excitement" to define arousal but excitement is an emotion that can be defined with a level of arousal. Hence, I would recommend using strength or intensity.

Line 96: There are emotion models with 15 emotions (Passion and Reason: Making Sense of Our Emotions, Richard and Bernice Lazarus, 1994) and 27 emotions (Self-report captures 27 distinct categories of emotion bridged by continuous gradients, Cowen and Keltner, 2017) which contradicts the claim that the existing emotion models are up to 8 emotions. I would recommend either changing it or limiting the context in order to claim that the max is 8 emotions.

Line 101: References are needed for related works that use Plutchik's emotion model

Line 121: WordNet-Affect is mentioned twice with different references.

Experimental design

The investigation carried out is extensive and explained clearly but there are a couple of shortcomings:

Clear research questions are needed for this study to highlight the contribution of the article. Is it the analysis of different models and approaches, the creation of a dataset, or both? For the analysis of different models, the models should be tested with additional datasets to see their behavior and for the creation of a dataset, descriptive statistics, exploratory data analysis, and data cleaning (e.g., debiasing and balancing) are necessary to produce validated datasets.

Why these linguistic features are utilized in the feature-based classification and not others is not clear. If this is according to Figure 1 (the information gain), why did the authors use linguistic features that are not represented in the information gain analysis, and why use the ones that do not carry informative features?

According to Table 1, the data is unbalanced in terms of individual emotions which the authors mentioned but the data is also skewed towards negative emotions overall (e.g., only 0.005% joy and 0.16% neutral). This would mean that the training (fine-tuning) of the models with this data would produce biased results. I appreciate that they used the "weighted average F1 score" but since the dataset is one of the contributions of this study, further effort should be put into eliminating or minimizing the skewness of the dataset if not the balance of individual labels.

Validity of the findings

The study lacks discussion/information about the connection between the emotions selected for the dataset and depression (or other mental disorders) since depression is heavily mentioned at the beginning of the article and the connection is never made between these emotions and depression.

The authors claim that there are significant improvements (e.g., line 431) but statistical tests supporting these claims are not presented. In order to claim significant differences, statistical data analysis should be performed (e.g., T-Test).

How can the proposed emotion dataset be compared and/or referred to the DAS (Depression, Anxiety, Stress) model? An investigation regarding the relation between the emotions introduced to the existing emotion models/datasets and the DAS model is required since the starting point for this work was to provide the model and datasets that would be utilized in mental health settings, especially the detection of depression through social media posts.

Additional comments

Below you can find a couple of typos I noticed and two suggestions (only suggestions for the authors regarding formatting)

Line 98: PLUTCHIK should be in lowercase after the capital letter as in "Plutchik" in the reference.
Line 104: What is ED? This is the first mention of ED.
Line 189-191: Duplicate phrase "could represent other states"
Line 204: "is" and "was" are both used one after the other.
Line 409-410: Duplicate phrase "ensure the output"

In order to present the models utilized in the analysis consistently, the RoBERTa-based models (301-306) should be presented in the same way BERT-based models (292-300) are presented (formatting only).

In Table 3, it would be better to present whether a model is a RoBERTa-based model or BERT-based model and whether it is a mono- or multilingual since the table has space to include this information and make the distinction between results clearer. Furthermore, I would suggest grouping the models based on which architecture they are based on and separating them with clear lines.

·

Basic reporting

This paper contributes to the field of emotion analysis in the Spanish language. The authors conducted a set of comparisons of multiple models, including state-of-the-art encoder-decoder transformers, generative large language models, and simple multilayer neural networks which leverage a set of linguistic features. The objective was to identify the one that performs best on the task of emotion classification. The contribution of the paper is two-fold: 1) the authors compiled several datasets from previous literature on emotion analysis in Spanish, creating a larger corpus that encompasses a total of 16 different emotions. This is an expansion from traditional datasets, which typically focus on the 6 emotions defined by Ekman; 2) the authors conducted extensive comparisons of large language models, both encoder and generative architectures, by fine-tuning them on the compiled corpus. They conclude that the MarIA encoder-based transformer achieves the best result on the task with a macro-F1 of 60%

Overall, the topic and content of the paper seem well-suited for the journal. The dataset compiled and curated by the authors can be of great value for future research in the field, particularly given that most available datasets are in English. The results from their experiments with different transformer-based architectures can serve as a benchmark to establish a baseline level of difficulty for the task.

More generally, the paper suffers several limitations that require revisions before being acceptable and considered for publication. Here are some suggestions to improve the paper:

1. I would highly recommend having a colleague, who is proficient in English and familiar with the research area, proofread your work. The manuscript would benefit from improved English, as in many places the writing style is unclear and often difficult to comprehend with several grammatical errors. For example, in lines:
- 12, which will -> which affected
- 49, to to -> to
- 65, in Section 6, presents -> Section 6 presents
- 79, As a results -> As a result
- 147, this presents up an -> this presents an
- 189-191, difficult to understand what you mean
- 204, This dataset is was -> This dataset was
- 221, 38.559 should be 38,559
- 249, be consistent with previous text: validation split instead of development
- 295, In terms of respect?
- 409, you are duplicating words
- Figure 6, did you mean confusion instead?

2. Enrich your state of the art/related work section. In the section on Natural Language Processing for Emotion Analysis, consider adding current research on emotion classification. In its current state, it reads more like a section highlighting common NLP techniques for general tasks rather than focusing specifically on emotion analysis:

- For example, in line 127, you mention that “Deep learning networks … have shown excellent performance in identifying emotions in text,” but you do not cite any relevant literature to substantiate this claim. Instead, some common methodologies used for embedding techniques in general are listed
In line 87, you say “researchers have made efforts to automate emotion analysis, but emotion detection remains a different task”. What is the difference between EA and ED? What makes ED more challenging?

- In line 121, you seem to have included two different references for the same methodology (WordNet-Affect). Please double check

- Double-check your references. Many of the papers you cite are from ArXiv. If there is a peer-reviewed version of such papers, make sure to cite those instead. Also, in line 129, you cite a paper from 2018, but it is dated 1802?

- In line 138, you mention “EA is a frequently studied topic with considerable amount of literature available,” then why not add references to such works? You cited the EmoEvalEs shared task, which is essentially the task that you focused on your paper. It would be good to mention how your contributions differ from this task. If it is only about testing LLMs on a larger corpus, then please specify

- In 166, you mention again that there are numerous works and shared tasks dedicated to the analysis of emotions, but you do not cite any

3. Double-check the structure of the manuscript, especially in Section 4 as it could be improved. For example, you have subsections (4.1 and 4.2) with very little information that could be merged together. Another suggestion would be to add a subsection that lays out the actual problem setup or definition. In the current state of your manuscript, it is difficult to tell if your problem is a multi-label or multi-class classification. After reading through Section 5, it becomes evident that it is the latter, but the paper would benefit from a more detailed description of your downstream task. Although you show the general pipeline in Figure 2, it would be helpful to expand more on each component, including a more formal presentation on what exactly the input is, what features are extracted, and what exactly is the output.

Experimental design

The evaluation is sound, and the findings are presented well. However, there are certain areas that need improvement:

1. The datasets section lacks details. One of the contributions of your manuscript is the compilation and curation of different datasets from literature to create a more comprehensive dataset for emotion detection in Spanish. Creating the dataset involved not only putting together relevant datasets but also several steps of curation such as text translation, annotations and relabeling. However, there is little information on these steps, particularly how the authors validated these annotations and translations. Were there any false positives within the already labeled datasets? Can you provide annotator agreement scores during the annotation process?

2. Another concern regarding datasets is related to the decision to split “long” documents and annotate their sentences individually. What do you consider a long document? Are individual sentences sufficient to express a particular emotion? Or do all these sentences get the same emotion label? Inferring a specific emotion for an individual sentence without the complete context could be very challenging and result in noisy examples. Also, while it is true that transformer models have limited contextual windows, there are alternative strategies to obtain a general representation of a long document. For example, you could have experimented with pooling the embeddings of separate sentences or considered other architectures, such as Longformers

3. The authors experimented with a feature combination approach called knowledge integration. What exactly does this strategy consider in terms of features? Is it a concatenation of sentence embeddings from a transformer-based model and the linguistic features in Section 4.3? This is not clear in text, a diagram or a more detailed description of this approach would make it clearer

4. In Section 5.1, you show results for the best set of hyperparameters obtained for each pre-trained model. What was the range of hyperparameters tested? A subsection that lays out the experimental setup including hyperparameters chosen would be a good idea

5. In Section 5.3, for choosing the best encoder-type model and encoder-decoder-type model, did you base your criteria on performance on the validation or test set? If these decisions are being made on the test set, then it is expected that there would be an improvement. All decisions should be made in the validation set, and if this was the case, it is not clear in the paper

Validity of the findings

All underlying data have been provided. Supplementary material and data have been provided for replication. Conclusions are sound and limited to supporting results

Additional comments

no comment

---

## Round 0.2 · Minor Revisions

The reviewers appreciate the effort invested in revisions. Minor improvements related to language and accuracy are still necessary.

·

Basic reporting

I would like to thank the authors for addressing all my previous comments in this section.

I would, on the other hand, encourage the authors to perform a second thorough English check (spelling, duplicate etc.)
For example (lines in the new manuscript):
Line 231 "given by each given by each"
Line 232 "that word to the of that word"

Experimental design

I would like to thank the authors for addressing all my previous comments in this section and for their responses.

Validity of the findings

I would like to thank the authors for addressing my previous comments in this section and for their responses.

I would like to, on the other hand, encourage the authors to further address the following points:

Regarding my previous comment "The study lacks discussion/information about the connection between the emotions selected for the dataset and depression (or other mental disorders) since depression is heavily mentioned at the beginning of the article and the connection is never made between these emotions and depression.":
I appraciate the new additions but it would be great to give a comment on which of the emotions are directly related to depression and how different models are performing at recognizing these emotions.


Regarding my previous comment "The authors claim that there are significant improvements but statistical tests supporting these claims are not presented. In order to claim significant differences, statistical data analysis should be performed (e.g., T-Test)."
I appraciate the explanation provided by the authors on the statistical analysis and why it is a limitation. On the other hand, the text still contains the same claim the the improvements are significant (line 496, line 534). I believe that in order to use the word "significant" for an outcome you have to perform the statistical analysis, otherwise there is no way of knowing whether the difference is significant or not. Hence, the authors should remove the word "significant" from these lines (496 and 534 in the new manuscript).

Additional comments

I would like to thank the authors for addressing all my previous comments in this section.

·

Basic reporting

Sufficient.

Experimental design

Sufficient.

Validity of the findings

Sufficient and proved.

Additional comments

I want to thank the authors for the effort made in their manuscript revision. The rebuttal letter was detailed and it addressed the concerns I had with the initial version of the paper. Overall, the paper is clear and meets the standards for the journal.

---

## Author Rebuttal · Round 0.2

# Comments from the editors and reviewers:

We would like to thank the editor and the reviewers for the time spent reviewing our paper and for the valuable comments they have provided, which have, in our opinion, helped us to improve the paper and clarify some points that were not well explained in the previous version.

Our point-to-point answers to the reviewers' and editor's comments are as follows.

## Reviewer 1: Yusuf Can Semerci

## Basic reporting

The article is written clearly in professional English but it needs to be checked for any typos and missing spaces between parentheses, words, and commas such as Line 154 and Line 104. The article is easy to follow and the raw data were shared. The methods utilized and the datasets used are explained clearly but there are a couple of shortcomings:

We thank the reviewer for his valuable comments. The manuscript has been revised and proofread.

The abstract lacks mention of NLP approaches, social media, and text modality. It would be clearer if these keywords were mentioned in the abstract as well.

Thanks to the reviewer for pointing this out. The abstract has been changed to add the following statement:

> *Social media platforms such as Twitter have become important hubs for health information as more people turn to these platforms to share their struggles and seek emotional support. Researchers have discovered a correlation between emotions and mental illnesses such as depression. This correlation provides a valuable opportunity for automated analysis of social media data to detect changes in mental health status that might otherwise go unnoticed, potentially preventing the development of more serious health consequences.*

Line 39: References are needed to support the claim that "there are remarkable results demonstrated with transformer-based models".

We have included references to support this statement.

Line 92: Arousal refers to the strength or the intensity of the emotion and excitement is an emotion. The authors used "level of excitement" to define arousal but excitement is an emotion that can be defined with a level of arousal. Hence, I would recommend using strength or intensity.

Thank you for the clarification. The document has been changed to reflect this.

Line 96: There are emotion models with 15 emotions (Passion and Reason: Making Sense of Our Emotions, Richard and Bernice Lazarus, 1994) and 27 emotions (Self-report captures 27 distinct categories of emotion bridged by continuous gradients,

Cowen and Keltner, 2017) which contradicts the claim that the existing emotion models are up to 8 emotions. I would recommend either changing it or limiting the context in order to claim that the max is 8 emotions.

Thank you for the clarification. The document has been changed to reflect this.

Line 101: References are needed for related works that use Plutchik's emotion model

Thanks to the reviewer for pointing this out. Related work on Plutchik's emotion model has now been added to the manuscript.

Line 121: WordNet-Affect is mentioned twice with different references.

Thanks to the reviewer for pointing this out. References have been checked.

## Experimental design

The investigation carried out is extensive and explained clearly but there are a couple of shortcomings:
Clear research questions are needed for this study to highlight the contribution of the article.

Thanks to the reviewer for pointing this out. The following RQs have been added to the Introduction:

- RQ1. What is the reliability of identifying negative fine-grained emotions?
- RQ2. What is the best approach to face the emotion analysis using the text as input?
- RQ3. Are generative models effective in identifying different emotions?

To accurately answer these RQs, we have expanded the article adding a comparison of individual emotions for the best models (Table 7) and added a final paragraph in Section 5 (Results and analysis), to indicate our findings after the analysis.

Is it the analysis of different models and approaches, the creation of a dataset, or both?

Thanks to the reviewer for pointing this out. We have added the following paragraph in the Introduction to clarify our contributions in EA.

> The paper makes significant contributions to the field of EA for the detection of mental disorders. 1) The dataset we compile and evaluate includes 16 different emotions using a multi-classification scheme. This method provides a unique approach by including emotions and states beyond those defined by Ekman's basic emotions. This dataset includes emotions such as loneliness, depression, suicidality, and hopelessness. 2) We evaluate this aforementioned dataset with several encoder-only, encoder-decoder, and feature integration models for text generation in EA due to the promising results reported in several studies.

For the analysis of different models, the models should be tested with additional datasets to see their behavior

Thanks to the reviewer for pointing this out. We have included the comparison with the EmoEvalES 2021 dataset in subsection 5.4. In this experiment, the encoder-only MarIA and the

encoder-decoder BLOOM-3b outperformed the best accuracy obtained by the GSI-UPM team (accuracy of 73.5507% and 73.1280% compared to 72.7657%)

and for the creation of a dataset, descriptive statistics, exploratory data analysis, and data cleaning (e.g., debiasing and balancing) are necessary to produce validated datasets.

We agree with the reviewer that the creation of a dataset requires descriptive statistics, exploratory data analysis and cleaning. However, we have already included in the description of the dataset (Section 3) the compilation and annotation process as well as an exploratory data analysis using the information gain of a set of linguistic features, to analyze how is the language used for each emotion (Figure 1).

With respect to data cleaning and debiasing, some details are added to the Section 3 as well as some examples of the compiled documents.

Why these linguistic features are utilized in the feature-based classification and not others is not clear. If this is according to Figure 1 (the information gain), why did the authors use linguistic features that are not represented in the information gain analysis, and why use the ones that do not carry informative features?

The linguistic features are used twice in the paper. The first use is to perform an exploratory analysis of the datasets by calculating the information gain (see Section 3). The second application is to use them as a baseline (see Table 3) to compare the performance of the LLMs. They are not included in the knowledge integration strategy because language models are able to learn information about the use of language in their attention mechanisms.

According to Table 1, the data is unbalanced in terms of individual emotions which the authors mentioned but the data is also skewed towards negative emotions overall (e.g., only 0.005% joy and 0.16% neutral). This would mean that the training (fine-tuning) of the models with this data would produce biased results. I appreciate that they used the "weighted average F1 score" but since the dataset is one of the contributions of this study, further effort should be put into eliminating or minimizing the skewness of the dataset if not the balance of individual labels.

We agree with the reviewer that the dataset is more skewed towards negative emotions. This is due to the nature of the datasets used to build the final corpus and the scope of our work, which focused on fine-grained emotions related to depression. However, we have included some research in Section 6 to address this weakness identified by the reviewer.

## Validity of the findings

The study lacks discussion/information about the connection between the emotions selected for the dataset and depression (or other mental disorders) since depression is heavily mentioned at the beginning of the article and the connection is never made between these emotions and depression.

We thank the reviewer for pointing this out. We have modified the paper to include more details on the relationship between the evaluated emotions from the classification reports of

the 2 best models (new Table 6). Below, we include the analysis of the emotions included in the paper.

> Table 6 shows the classification reports for the best fine-grained model, MarIA, and the best generative model, mean-based ensemble learning. The results show that both strategies are similar for fine-grained emotion analysis. The big difference is observed in the joy emotion, where Maria achieves an F1 score of 42.2535%, while the mean-based ensemble learning approach achieves only 31.250%, outperforming the results in both precision and recall. It is also observed that the ensemble model outperforms MarIA on several negative emotions such as depressed, disappointment, embarrassment, grief, nervousness}, sadness, and suicidal. In terms of precision and recall, both strategies show similar behavior since both metrics are similar. However, ensemble learning based on the mean shows a larger difference between precision and recall for the identification of texts labeled joy. Also, neither strategy was able to identify the only instance of surprise in the test split.

The authors claim that there are significant improvements (e.g., line 431) but statistical tests supporting these claims are not presented. In order to claim significant differences, statistical data analysis should be performed (e.g., T-Test).

To compare the models using statistical tests we should have performed an evaluation using nested cross-validation instead of just using the validation test during hyperparameter tuning. However, due to the large number of evaluated language models and the lack of hardware resources, this type of validation is currently not possible. Moreover, since we use Bayesian optimization to select the next set of hyperparameters during parameter optimization, we cannot guarantee that the same set of parameters will be analyzed in the folds, so we should change our pipeline to perform a fair statistical test.

However, we add this limitation to the conclusions and include it as future work.

> As a limitation, statistical tests should be conducted to perform a better comparison of the models, since using a fixed custom validation split can bias some decisions about the best performing models. In this sense, we propose to extend this work using nested cross-validation for a better comparison.

How can the proposed emotion dataset be compared and/or referred to the DAS (Depression, Anxiety, Stress) model? An investigation regarding the relation between the emotions introduced to the existing emotion models/datasets and the DAS model is required since the starting point for this work was to provide the model and datasets that would be utilized in mental health settings, especially the detection of depression through social media posts.

The following paragraph has been added to the Introduction section.

> Unlike from other approaches to mental health detection, such as DAS (Depression, Anxiety, Stress) detection models, emotion analysis models can provide greater consistency and enable the visualization of mood changes through published text, thereby avoiding false positives. For example, a semi-supervised machine learning model, DASentimental, has been proposed in \citep{bdcc5040077} to extract depression, anxiety, and stress from written text. However, it is only capable of identifying negative emotions.

## Additional comments

Below you can find a couple of typos I noticed and two suggestions (only suggestions for the authors regarding formatting)

Line 98: PLUTCHIK should be in lowercase after the capital letter as in "Plutchik" in the reference. Solved
Line 104: What is ED? This is the first mention of ED.
Line 189-191: Duplicate phrase "could represent other states"
Line 204: "is" and "was" are both used one after the other.
Line 409-410: Duplicate phrase "ensure the output"

The document has been edited to correct typographical, spelling, and formatting errors.

In order to present the models utilized in the analysis consistently, the RoBERTa-based models (301-306) should be presented in the same way BERT-based models (292-300) are presented (formatting only).

The details have been added to the article.

In Table 3, it would be better to present whether a model is a RoBERTa-based model or BERT-based model and whether it is a mono- or multilingual since the table has space to include this information and make the distinction between results clearer. Furthermore, I would suggest grouping the models based on which architecture they are based on and separating them with clear lines.

Table 3 has been updated accordingly.

Reviewer: Kin Ng

## Basic reporting

This paper contributes to the field of emotion analysis in the Spanish language. The authors conducted a set of comparisons of multiple models, including state-of-the-art encoder-decoder transformers, generative large language models, and simple multilayer neural networks which leverage a set of linguistic features. The objective was to identify the one that performs best on the task of emotion classification. The contribution of the paper is two-fold: 1) the authors compiled several datasets from previous literature on emotion analysis in Spanish, creating a larger corpus that encompasses a total of 16 different emotions. This is an expansion from traditional datasets, which typically focus on the 6 emotions defined by Ekman; 2) the authors conducted extensive comparisons of large language models, both encoder and generative architectures, by fine-tuning them on the compiled corpus. They conclude that the MarIA encoder-based transformer achieves the best result on the task with a macro-F1 of 60%

Thanks to the reviewer for these encouraging comments.

Overall, the topic and content of the paper seem well-suited for the journal. The dataset compiled and curated by the authors can be of great value for future research in the field, particularly given that most available datasets are in English. The results from their experiments with different transformer-based architectures can serve as a benchmark to establish a baseline level of difficulty for the task.

Thanks for the reviewer for these valuable comments.

More generally, the paper suffers several limitations that require revisions before being acceptable and considered for publication. Here are some suggestions to improve the paper:
I would highly recommend having a colleague, who is proficient in English and familiar with the research area, proofread your work. The manuscript would benefit from improved English, as in many places the writing style is unclear and often difficult to comprehend with several grammatical errors. For example, in lines:

- 12, which will -> which affected.
- 49, to to -> to.
- 65, in Section 6, presents -> Section 6 presents.
- 79, As a results -> As a result.
- 147, this presents up an -> this presents an.
- 189-191, difficult to understand what you mean.
- 204, This dataset is was -> This dataset was.
- 221, 38.559 should be 38,559.
- 249, be consistent with previous text: validation split instead of development.
- 295, In terms of respect?
- 409, you are duplicating words.

- Figure 6, did you mean confusion instead?

The document has been edited to correct typographical, spelling, and formatting errors.

Enrich your state of the art/related work section. In the section on Natural Language Processing for Emotion Analysis, consider adding current research on emotion classification. In its current state, it reads more like a section highlighting common NLP techniques for general tasks rather than focusing specifically on emotion analysis:

Thanks to the reviewer for pointing this out. Section 2 has been expanded to include recent work on EA.

For example, in line 127, you mention that "Deep learning networks … have shown excellent performance in identifying emotions in text," but you do not cite any relevant literature to substantiate this claim. Instead, some common methodologies used for embedding techniques in general are listed.

Thanks to the reviewer for pointing this out. References have been revised and related work on other EA taxonomies has been added.

In line 87, you say "researchers have made efforts to automate emotion analysis, but emotion detection remains a different task". What is the difference between EA and ED? What makes ED more challenging?

These terms are synonymous, but we have normalized their appearance in the document for clarity.

In line 121, you seem to have included two different references for the same methodology (WordNet-Affect). Please double check

This issue has been resolved.

Double-check your references. Many of the papers you cite are from ArXiv. If there is a peer-reviewed version of such papers, make sure to cite those instead. Also, in line 129, you cite a paper from 2018, but it is dated 1802?

All references have been reviewed and checked for new peer-reviewed versions.

In line 138, you mention "EA is a frequently studied topic with considerable amount of literature available," then why not add references to such works? You cited the EmoEvalEs shared task, which is essentially the task that you focused on your paper. It would be good to mention how your contributions differ from this task. If it is only about testing LLMs on a larger corpus, then please specify.

The state of the art has been improved according to this suggestion.

We consider, however, that our contributions are now clearer with the following paragraph in the Introduction:

> The paper makes significant contributions to the field of EA for the detection of mental disorders. 1) The dataset we compile and evaluate includes 16 different emotions using a multi-classification scheme. This method provides a unique approach by including

emotions and states beyond those defined by Ekman's basic emotions. This dataset includes emotions such as loneliness, depression, suicidality, and hopelessness. 2) We evaluate this aforementioned dataset with several encoder-only, encoder-decoder, and feature integration models for text generation in EA due to the promising results reported in several studies.

It is worth noting that we have included in the manuscript a comparison with the EmoEval 2021 dataset using our pipeline.

In 166, you mention again that there are numerous works and shared tasks dedicated to the analysis of emotions, but you do not cite any

The state of the art has been improved according to this suggestion.

Double-check the structure of the manuscript, especially in Section 4 as it could be improved. For example, you have subsections (4.1 and 4.2) with very little information that could be merged together.

We find this a very interesting suggestion, because in reviewing the article, we realized that an improvement in structure was needed, as suggested by the reviewer. We have reorganized this section with several changes that we hope will improve the compression of the article. First, we have merged sections 4.1 and 4.2. We've added a specific section to describe the baseline, and we've renamed our strategies to "encoder-only", "encoder-decoder", and "feature integration".

Another suggestion would be to add a subsection that lays out the actual problem setup or definition. In the current state of your manuscript, it is difficult to tell if your problem is a multi-label or multi-class classification. After reading through Section 5, it becomes evident that it is the latter, but the paper would benefit from a more detailed description of your downstream task.

In the introduction, we have added some details that the dataset is prepared for performing a multi-class document classification.

Although you show the general pipeline in Figure 2, it would be helpful to expand more on each component, including a more formal presentation on what exactly the input is, what features are extracted, and what exactly is the output.

The figure has been modified according to our structure in the Materials and Methods section.

## Experimental design

The evaluation is sound, and the findings are presented well. However, there are certain areas that need improvement:
The datasets section lacks details. One of the contributions of your manuscript is the compilation and curation of different datasets from literature to create a more comprehensive dataset for emotion detection in Spanish. Creating the dataset involved not only putting together relevant datasets but also several steps of curation such as text translation, annotations and relabeling. However, there is little information on

these steps, particularly how the authors validated these annotations and translations. Were there any false positives within the already labeled datasets? Can you provide annotator agreement scores during the annotation process?

Additional details about the dataset annotation and relabeling have been added to the article, as well as some examples of the compiled documents. However, it is not possible for us to add information about the agreement between the annotators, since the differences between the labels of the subset of the dataset that we relabeled by hand were resolved in different meetings. It should be noted that this relabeling was only done on a subset of the dataset, since we decided to split some of the long documents into smaller sentences.

Another concern regarding datasets is related to the decision to split "long" documents and annotate their sentences individually. What do you consider a long document? Are individual sentences sufficient to express a particular emotion? Or do all these sentences get the same emotion label? Inferring a specific emotion for an individual sentence without the complete context could be very challenging and result in noisy examples. Also, while it is true that transformer models have limited contextual windows, there are alternative strategies to obtain a general representation of a long document. For example, you could have experimented with pooling the embeddings of separate sentences or considered other architectures, such as Longformers.

We would like to thank the reviewer for this interesting comment.

The decision to split long documents was based on the maximum limit of transformers with some documents collected in discussion forums (rather than microblogging). We did not assign the same emotion because, as the reviewer comments, it is not possible to assume that subsets of a document contain the same emotion as the sentence in its entire context, so a relabeling process was performed in which parts of the sentence whose emotion could not be decided were eliminated.

As for other strategies for evaluating long texts, we have not yet been able to evaluate solutions such as longformers, but we consider this an interesting avenue for evaluation.

The authors experimented with a feature combination approach called knowledge integration. What exactly does this strategy consider in terms of features? Is it a concatenation of sentence embeddings from a transformer-based model and the linguistic features in Section 4.3? This is not clear in text, a diagram or a more detailed description of this approach would make it clearer.

The knowledge integration strategy consists of training a new multi-input neural network from the embeddings of the remaining models. This strategy only considers encoder-only models, since the linguistic features were only used to describe the data set and as a baseline. Therefore, we modified the description of this strategy in the paper as follows: "The second approach to feature combination is called knowledge integration. This method involves training a multi-input neural network from scratch, incorporating all the sentence embeddings for each encoder-only mode". (see Section 4.3.)

4. In Section 5.1, you show results for the best set of hyperparameters obtained for each pre-trained model. What was the range of hyperparameters tested? A subsection

that lays out the experimental setup including hyperparameters chosen would be a good idea

We have added this information to the manuscript:

> The hyperparameters under consideration, along with their respective interval ranges, are: (1) weight decay (ranging from 0 to 0.3), (2) training lot size (ranging from 8 to 16), (3) number of training epochs (ranging from 1 to 6), and (4) learning rate (ranging from 1e-5 to 5e-5).

5. In Section 5.3, for choosing the best encoder-type model and encoder-decoder-type model, did you base your criteria on performance on the validation or test set? If these decisions are being made on the test set, then it is expected that there would be an improvement. All decisions should be made in the validation set, and if this was the case, it is not clear in the paper

These decisions are being made with the custom validation split. We have clarified this on the paper on Sections 5.1 and 5.2.

## Validity of the findings

All underlying data have been provided. Supplementary material and data have been provided for replication. Conclusions are sound and limited to supporting results

We thank the reviewer for the encouraging comments.

## Additional comments

no comment

---

## Round 0.3 · Minor Revisions

The paper is not carefully edited, despite the claims of the authors. A somewhat random check led me to discover that two references are in fact to the same paper (Brown 2020a and Brown 20202b). But what puzzles me is the claim that LLMs understand language* (even within quotes) which I find nowhere in Brown et al. I would recommend the authors to use scientific rigour, as there is sufficient journalistic non-sense on this topic, and to carefully state the facts, especially when attributed to other authors.

From the paper:
* 368-369 According to Brown et al. (2020a), LLMs have the ability to learn from few or even no examples due to their inherent ability to “understand” language.

---

## Round 0.4 · accepted · Accept

This paper is ready for publication in its current form.

---

## Author Rebuttal · Round 0.4

# Comments from the editors and reviewers:

We would like to thank the editor and the reviewers for the time they spent reviewing our paper and for the valuable comments they provided, which we believe helped us improve the paper and clarify some points that were not well explained in the previous version.

The editor informed us twice that there are some problems with the English language and some references, and therefore the language and the manuscript need to be improved. Accordingly, we first sought the help of a colleague who is an Associate Professor of English Linguistics and who is familiar with the subject matter of this manuscript, and we revised and updated the manuscript. Next, we conducted another careful review to check the entire manuscript, references, and direct quotations.

Our point-by-point responses to the reviewers' and editor's comments are as follows.

## Reviewer 1: Yusuf Can Semerci

## Basic reporting

I would like to thank the authors for addressing all my previous comments in this section.

I would, on the other hand, encourage the authors to perform a second thorough English check (spelling, duplicate etc.)

For example (lines in the new manuscript):

Line 231 "given by each given by each"

Line 232 "that word to the of that word"

We thank the reviewer for his valuable comments. The manuscript has been revised and proofread.

## Validity of the findings

Regarding my previous comment "The study lacks discussion/information about the connection between the emotions selected for the dataset and depression (or other mental disorders) since depression is heavily mentioned at the beginning of the article and the connection is never made between these emotions and depression.":

I appraciate the new additions but it would be great to give a comment on which of the emotions are directly related to depression and how different models are performing at recognizing these emotions.

We thank the reviewer for his valuable comments. We believe that his recommendations have improved the overall quality of the paper.

Regarding my previous comment "The authors claim that there are significant improvements but statistical tests supporting these claims are not presented. In order to claim significant differences, statistical data analysis should be performed (e.g., T-Test)."
I appraciate the explanation provided by the authors on the statistical analysis and why it is a limitation. On the other hand, the text still contains the same claim the the improvements are significant (line 496, line 534). I believe that in order to use the word "significant" for an outcome

you have to perform the statistical analysis, otherwise there is no way of knowing whether the difference is significant or not. Hence, the authors should remove the word "significant" from these lines (496 and 534 in the new manuscript).

Thanks to the reviewer for pointing this out. We have deleted the word "significant" from lines 496 and 534.

## Reviewer 2 (Kin Wai Ng Lugo)

### Basic reporting

*Sufficient.*

### Experimental design

*Sufficient.*

### Validity of the findings

*Sufficient and proved.*

### Additional comments

*I want to thank the authors for the effort made in their manuscript revision. The rebuttal letter was detailed and it addressed the concerns I had with the initial version of the paper. Overall, the paper is clear and meets the standards for the journal*

We thank the reviewer for his valuable comments. We believe that his recommendations have improved the overall quality of the paper.